# Motion Characteristics of Gas–Liquid Two-Phase Flow of Microbubbles in a Labyrinth Channel Used for Aerated Drip Irrigation

Yanfang Liu [1], Guocui Wang [1], Xianna Zhang [1], Hongchen Li [1,*], Bingcheng Si [1,2,*], Wenqian Liu [1] and Zhenhua Zhang [1]

1    College of Resources and Environment Engineering, Ludong University, Yantai 264025, China
2    Department of Soil Science, University of Saskatchewan, Saskatoon, SK S7N 5A8, Canada
*    Correspondence: lihc@ldu.edu.cn (H.L.); bingchengsi@sina.com (B.S.);
     Tel.: +86-188-6568-0862 (H.L.); +86-178-5457-9063 (B.S.)

**Abstract:** The indefinite characteristics of gas–liquid two-phase flow limit the usage of aerated drip irrigation. Gas–liquid two-phase flow in a labyrinth channel was observed using a particle tracking velocimetry (PTV) technique in this study. The motion trajectory and velocity vector of large numbers of microbubbles were characterized and analyzed at 0.01, 0.02, 0.04 MPa inlet pressure and in three labyrinth channels with different geometries. The results indicated that bubbly flow was the typical flow pattern in a labyrinth channel, with slug flow occurring occasionally. Smooth and gliding motion trajectories of bubbles were observed in the mainstream zone, while twisted trajectories were seen in the vortex zone. Increasing the inlet pressure increased the number of bubbles and the trajectory length in the vortex zone. When the inlet pressure increased from 0.02 to 0.04 MPa, the 25th percentile of Rc-t (the Ratio of Circular path length in the vortex zone to the Total trajectory length for a single bubble) increased from 0 to 12.3%, 0 to 6.1%, and 0 to 5.2% for channels A, B, and C, respectively; the 75th percentile increased from 31.3% to 43.9%, 27.5% to 31.9%, and 18.7% to 22.3%. The velocity vectors of the bubbles showed position dependence. Bubbles with high speed were found in the mainstream zone with their directions parallel to the water flow direction. Bubbles with low speed were seen in the vortex zone, moving in all directions. With inlet pressure increased from 0.01 to 0.04 MPa, the mean instantaneous velocities of bubbles in channels A, B, and C are increased by 106.2%, 107.6%, and 116.6%, respectively. At 0.04 MPa, channel A has the longest path length and the highest instantaneous velocity of bubbles in the vortex zone among three channels, exhibiting the highest anti-clogging performance of the three channels. This study will help in the comprehensive understanding of gas–liquid two-phase flow in a labyrinth channel used for aerated drip irrigation.

**Keywords:** aerated drip irrigation; labyrinth channel; gas–liquid two-phase flow; microbubbles; motion characteristics

## 1. Introduction

Aerated drip irrigation (ADI) is a form of drip or subsurface drip irrigation which uses gas–liquid two-phase flow to deliver water and nutrients to crop root zones. Such a system can reduce the hypoxia of soil, whilst ensuring a good growth environment for roots, and improving water-use efficiency, along with crop yield and quality [1–3]. Generally, such a system uses an air compressor and a Venturi intake system or a micro-nanobubble generating device to add oxygen-containing gas into the irrigation water. The gas exists in the water in the form of bubbles with diameters in the nanometer to centimeter range [4–6]. Therefore, the fluid in the pipe and the emitter flow channel can be considered to move as a gas–liquid two-phase flow. The flow characteristics of the two-phase fluid in the emitter channel are very different from those of the single-phase flow of water, due to the interaction between the air bubbles and the water during the transportation process. This interaction will greatly affect the hydraulic performance and anti-clogging performance of

the emitter and has a significant influence on the operational state of the drip irrigation system [6–8].

At present, research into ADI is mainly focused on the positive effects it has on soil physical structure, soil chemical composition, soil microbial activities and crop growth, as well as its influences upon the utilization efficiency of water, nutrients, and greenhouse gas emissions [9–16]. There have also been many studies concerning the transport process of water–gas two-phase flow in pipeline systems. Torabi et al. explored the effect of various factors on bubble movement and the emitter air flow-rate distribution in recirculating drip irrigation systems. They showed that the two-phase flow of air and water in drip irrigation pipes is a complex process. The availability and supply of air bubbles to the system was broadly determined by connector geometry, pipe diameter, and emitter flow rate [17]. Lei et al. and Torabi et al. proposed a method of increasing the existence time and distribution uniformity of dissolved oxygen in drip irrigation systems by improving the pipe layout and adding a surfactant [18–21]. Bhattarai et al. studied the changes in the shape and size of the bubbles, the air void fraction along the drip line, and the influences of surfactant and orientation of the emitter to the ground plane. They found that when more surfactant was used in the emitter down position, a greater air fraction moved further along the drip line [22,23]. In addition, Li et al. suggested that micro-nanobubble water could improve the anti-blocking performance of drip irrigation systems [6]. They studied the clogging process of a system using micro-nanobubble water for drip irrigation. The emitter is the key component of a drip irrigation system and it commonly has a labyrinth structure. Presently, there appear to have been no studies of gas–liquid two-phase flow and micro-nanobubbles in the labyrinth channel of aerated drip irrigation systems.

Researchers often use a high-speed digital camera system to capture images of the labyrinth channel, due to its complex structure and small size. Li et al. used DPIV measurement technology to measure the water flow field in emitters and to carry out two-dimensional non-disturbance tests within the flow channels of emitters. Results clearly showed the velocity distribution at pressures between 10 and 150 kPa using DPIV measurements [24]. Wei et al. chose silica sand as the solid phase in an experiment that used a micro-PIV technique to capture the flow of water containing suspended sand particles inside a labyrinth channel at pressures ranging from 40 to 150 kPa. The velocity vectors of the silica sand particles were obtained by processing the data with software [25]. Similarly, Yu et al. observed the solid–liquid two-phase flow using a PTV technique. The movement of ten grains of sand with Stokes numbers from 5 to 10 was analyzed, including the mean speed and the mean running time [26]. Existing research into the flow characteristics of labyrinth channels using imaging technology is primarily focused on single-phase flow (water) and solid–liquid two-phase flow (water and sand) [27,28]. However, research into gas–liquid two-phase flow in microchannels is more common in chemical and microfluidic fields.

The mechanism of, and factors influencing, gas–liquid two-phase flow have been studied using high-speed microphotography and digital image processing technology in microchannels of sizes ranging from 1 micron to 1 millimeter [29–31]. Venkatesan et al. investigated gas–liquid two-phase flow in a horizontal circular pipe with an inner diameter that varied between approximately 0.6 mm and 3.4 mm. The flow patterns seen for horizontal flows were stratified smooth, dispersed bubble, slug, and annular. The influence of pipe diameter on flow pattern was also observed [32]. Saisorn et al. showed that gas–liquid two-phase flow was mainly affected by surface tension, gas–liquid viscosity, and inertia, while gravity had little influence [33]. Zhang et al. described how the gas–liquid two-phase flow in a microchannel has different characteristics than the flow at a conventional scale. As the pipe diameter decreases, the effect of gravity on bubble movement reduces and the effect of surface tension increases. Pressure also plays an important role in gas–liquid two-phase flow [34]. The same flow pattern will exhibit different characteristics at different pressures [35]. These studies investigated the formation process and mechanism of microbubbles, the flow pattern of two-phase flow (and the

influence of the former on the latter), with the aim of understanding how to control the microbubble generation process in microchannels. Consequently, the research discussed is of limited reference value to this work since it had a different objective and was carried out in a different experimental environment.

In this study, gas–liquid two-phase flow was observed using a PTV technique in a labyrinth channel with micro-nanobubble water to explore the effects of inlet pressure and the geometry of the labyrinth channel on bubble movement and the patterns of gas–liquid two-phase flow. This study provides data for future studies of hydraulic performance and anti-clogging performance of aerated drip irrigation systems and will be helpful for numerical simulations of gas–liquid two-phase flow in labyrinth channels.

## 2. Materials and Methods

### 2.1. Micro-Nanobubble Water

The micro-nanobubble water used in the experiment was made using a Series Fine Bubble Injector (LF1500; Shanghai Xingheng Technology Co., Ltd., Shanghai, China). The equipment produces micron-sized and nanometer-sized bubble water evenly and continuously and has adjustable water supply pressure and an adjustable gas–water ratio, ranging from 0% to 10%. Half of the micron-sized bubbles have a diameter of 30–40μm, while half of the nanometer-sized bubbles have a diameter of 160–170nm. The indexes of nanoscale bubbles in the mixture at room temperature are shown in Table 1. The micro-nanobubble water used in this experiment contained numerous invisible nanoscale bubbles. The gas–water ratio of micro-nanobubbles water used was 10%.

**Table 1.** Indexes of nanoscale bubbles in micro-nanobubble water with a gas–water ratio of 10% at room temperature.

| Indexes | Value |
|---|---|
| Average bubble diameter (nm) | 200.2 |
| Median bubble diameter (nm) | 155.2 |
| AD10 (nm) | 108.0 |
| D50 (nm) | 164.1 |
| D90 (nm) | 297.8 |
| Concentration (particles/mL) | $3.44 \times 10^8$ |

Note: D10 means that 10% of the bubbles are smaller than this diameter. D50 and D90 have similar meanings.

### 2.2. Labyrinth Channels and Their Pressure-Flow Relationship

Three common labyrinth channels (marked A, B, and C) were chosen for this experiment; their geometry and dimensions are shown in Figure 1. Each labyrinth channel was composed of 24 structural units of 2.0 mm length. The widths of the minimum flow sections were 0.8660, 0.8000, and 0.8660 mm, respectively, for channels A, B, and C. Each channel was 1.0 mm deep.

Labyrinth channel models were made using three transparent polymethylmethacrylate (PMMA) plates, for ease of observation. Each labyrinth channel was milled on a 1 mm plate, both sides of which were covered with 3 mm plates to seal the channel. Ten bolts were used to secure the three Perspex plates, and the model was then sealed with waterproof adhesive (Figure 2).

The flow quantity–pressure relationship (*q-H* relationship) of the three labyrinth channels was tested at non-aeration conditions, with the pressure ranging from approximately 0.02 to 0.15 MPa. The results are shown in Table 2.

**Table 2.** *q-H* relationship of the three labyrinth channels at non-aeration conditions.

| Labyrinth Channel | *q-H* Relationship |
|---|---|
| A | $q = 6.2554H^{0.4734}$ |
| B | $q = 6.9107H^{0.4705}$ |
| C | $q = 8.2504H^{0.4988}$ |

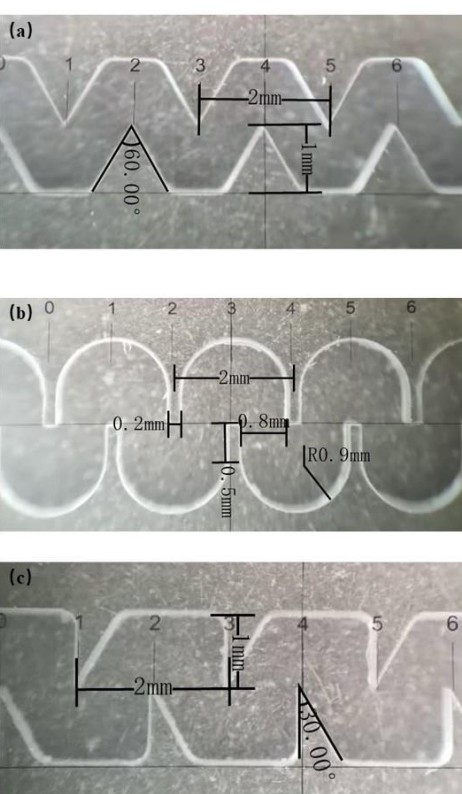

**Figure 1.** Geometry and dimensions of (**a**) labyrinth channel A, (**b**) labyrinth channel B, and (**c**) labyrinth channel C.

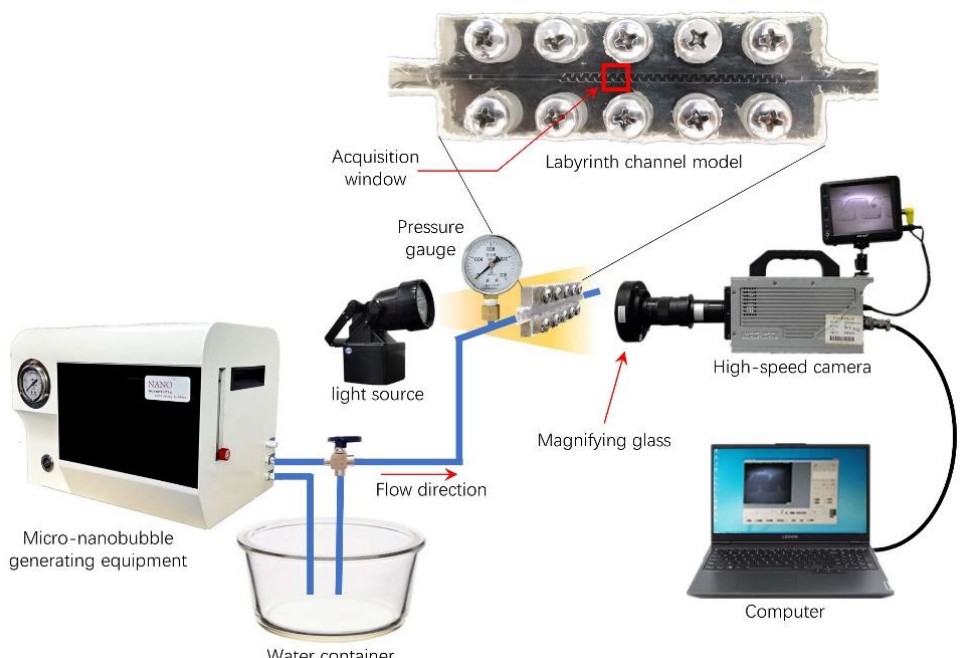

**Figure 2.** Experimental set-up.

*2.3. Aerated Drip Irrigation System*

A Series Fine Bubble Injector provided micro-nanobubble pressurized water for a drip irrigation system. A transparent PU pipe with an inner diameter of 6 mm was used to provide the water to the labyrinth channel model. The length of the pipe, from the

water supply outlet to the labyrinth channel model, was about 40.3 cm. The pressure at the inlet of the labyrinth channel was adjusted using the valve on the pipeline, and a pressure gauge (minimum scale value of 10 kPa) was installed at the inlet of the labyrinth channel to monitor the inlet pressure (Figure 2). Before the experiment, the pipes and the labyrinth channels were flushed with clean water for 24 h.

### 2.4. Imaging of the Labyrinth Channels

### 2.4.1. Equipment

The imaging equipment consisted of a continuous light source, a high-speed digital camera system (Hotshot 512SC; Nac Image Technology, Tokyo, Japan), and a magnifying glass VS-M0910 (Weishi Digital Image Technology Co., Ltd., Xi'an, China) (Figure 2). The gas–liquid two-phase flow through the fifth and sixth structural units (located at the front of the flow channel) was filmed during the test at between approximately 2000 and 4000 frames per second. The shooting field was about 2.5 structural units in length to ensure that 2 complete structural units could be seen in the pictures.

### 2.4.2. Image Processing

The PTV images were analyzed using IMAGE PRO PLUS 6.0. The trajectory of the target microbubble in the labyrinth channel under observation was plotted by tagging its position in successive frames. The displacement of the target microbubble during the capture of any two consecutive frames was obtained by tracking the position of the target bubble on these two successive frames. The bubbles in two consecutive frames were identified by eyeballing the size of bubbles and by the consistency of their motion. The lengths of bubble trajectory and displacement were measured using Image J v1.52u. The value of the velocity vector was calculated by dividing the displacement length by the time interval between two consecutive frames. The direction of the velocity vector was indicated by joining the two positions on the two consecutive frames.

### 2.5. Test Design and Data Processing

In this experiment, a total of nine treatments were designed, with three different labyrinth channel geometries (A, B, C) and three inlet pressures of 0.01 (V1), 0.02 (V2), and 0.04 (V3) MPa.

For each treatment, three videos were shot at 30 min, 60 min, and 90 minof the drip irrigation system operation, forming three replicates of the experiment to reduce the test error caused by system instability. Therefore, there are 27 videos in total.

For each video, we sampled about 80–100 bubbles to analyze bubble velocity vectors and 20–25 bubbles to analyze bubble trajectories. The bubbles were manually randomly selected during the video playback.

The data obtained in the experiment were analyzed statistically using SPSS 22.0.

## 3. Results

### 3.1. Flow Pattern of Gas-Liquid Two-Phase in a Labyrinth Channel

Three typical phenomena captured during the experiment are shown in Figure 3. Both bubbly flow and slug flow were observed in all nine treatments (Figure 3a,b), with slug flow occurring only occasionally.

When there was bubbly flow, the distribution of bubbles in the flow passage was relatively uniform, with round bubbles of a diameter between 10 and 100 μm, approximately (Figure 3a). Bubble coalescence and conglutination to the inner wall were observed in the test but bubble breakage was not observed. It should be noted that the images show the results of the three-dimensional fluid superimposed onto the two-dimensional plane, since the labyrinth channel is 1.0 mm deep. Thus, the bubbles that appear to be clustered together in the picture may not actually be so.

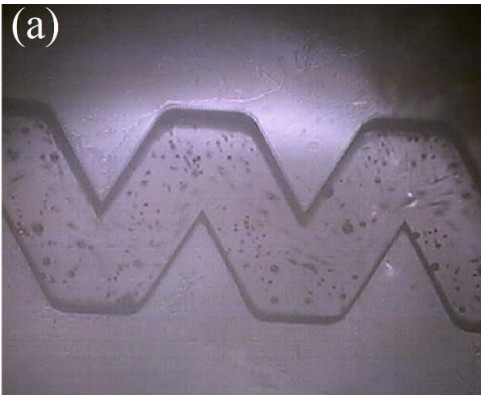 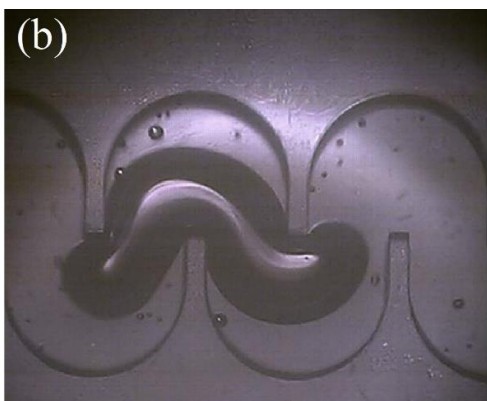

**Figure 3.** Gas–liquid two-phase flow pattern of (**a**) bubbly flow (AV2) and (**b**) slug flow (BV1).

Occasionally, bubbles with diameters between approximately 1 and 2 mm entered the labyrinth channel due to the coalescence of bubbles in the silicone hose that transported the bubbles and water before the labyrinth structure. When several consecutive millimeter-sized bubbles passed through the labyrinth channel, a slug flow occurred (Figure 3b). At this point, the millimeter bubble, deformed severely by the labyrinth channel geometry, almost filled the entire cross section of the labyrinth channel, with its length reaching approximately 2 to 3 millimeters. Even so, these bubbles could pass through the labyrinth channel in the field of vision completely, with no splitting or breaking observed during the whole process. However, a millimeter-sized bubble subsumed micron-sized bubbles sticking to the side wall as it moved along, and then carried them out of the labyrinth channel. Consequently, to a certain extent, the slug flow created a scouring action along the whole labyrinth passage. This phenomenon was also observed by Zhu et al. [36]. They found that large bubbles absorbed and swept away the microbubbles on the inner wall of a rectangular microchannel (2 mm × 1.8 mm).

*3.2. Bubble Trajectory*

Most of the time, there was a bubbly flow pattern in the labyrinth channel. Therefore, we studied the motion characteristics of micron-sized bubbles in the bubbly flow. Figure 4 shows an example of bubble trajectories in three labyrinth channels at three different inlet pressures (each color represents one bubble). The bubble trajectory near the center line of the labyrinth channel (the mainstream zone) is smooth and gliding, while the bubble trajectory near the groove of the labyrinth channel (the vortex zone) is chaotic and twisted. Observing a single bubble trajectory, the movement has an element of randomness to it. After entering the labyrinth channel in the field of vision, the bubbles either passed directly along the center line of the labyrinth channel or entered one or more grooves and took a circular route there. Then, they returned to the mainstream region after one or more irregular circular path lines. At the vortex zone, the diameter of the circular trajectory is large and its shape is regular at low pressure (Figure 4a), while the opposite is true at high pressure (Figure 4c).

In labyrinth flow channels, solid particles tend to settle in the vortex zone [37–39]. Therefore, the longer the trajectory in the vortex zone, the greater the probability of bubble attachment to particles, which is conducive to sediment removal. Therefore, the ratio of the circular path line of a single bubble to its total length is an effective index that reflects the bubble's cleaning efficiency. Therefore, we defined a new index named Rc-t (the Ratio of Circular path length in vortex zone to the Total trajectory length for a single bubble). The Rc-t values for all the bubble samples from the nine treatments (approximately sixty–seventy-five samples per treatment) were statistically analyzed and depicted in a violin plot (Figure 5), with significance analysis conducted.

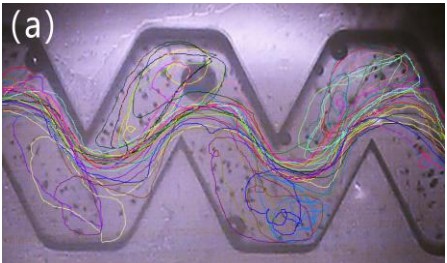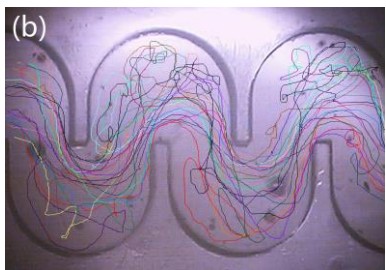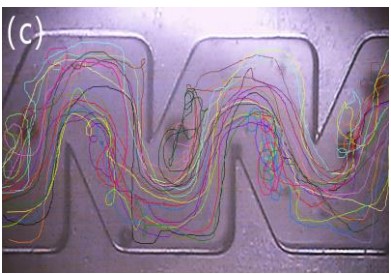

**Figure 4.** An example of bubble trajectories of (**a**) AV1, (**b**) BV2, and (**c**) CV3. Each colored line represents the trajectory of a bubble.

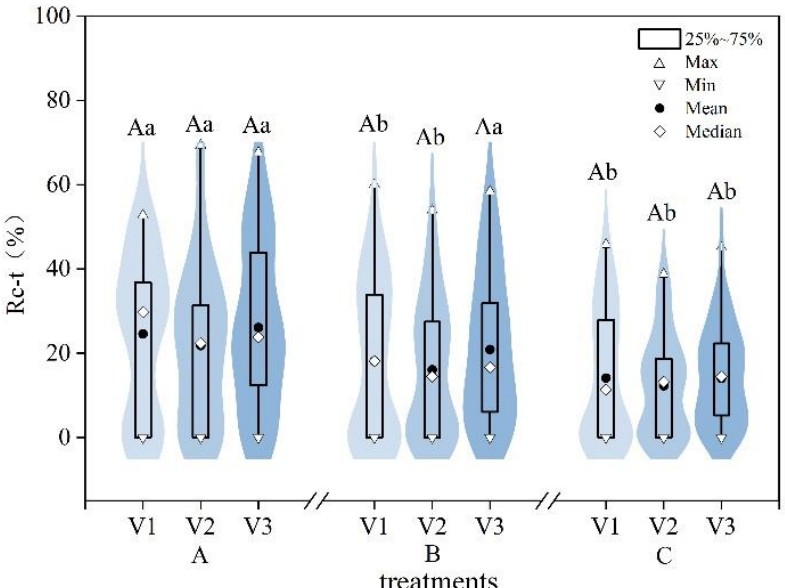

**Figure 5.** A violin plot of Rc-t values in nine treatments. Different uppercase letters indicate a significant difference among different channels at the same inlet pressure; different lowercase letters indicate significant difference among different inlet pressures for the same labyrinth channel ($p < 0.05$).

The movement of bubbles is highly stochastic, meaning it is necessary to focus on the behavior of the bubble population rather than individual bubbles. As shown in Figure 5, the mean value of Rc-t in the same labyrinth channel is about 20%, with no significant differences between the three pressures. When the inlet pressure increased from 0.02 to 0.04 MPa, the 25th percentile of Rc-t increased from 0 to 12.3%, 0 to 6.1%, and 0 to 5.2% for channels A, B, and C, respectively; the 75th percentile increased from 31.3% to 43.9%, 27.5% to 31.9%, and 18.7% to 22.3%. The increasing inlet pressure resulted in a greater number of bubbles entering the vortex zone, and increased path lengths. However, Rc-t at 0.01 MPa(V1) is very different, with its 75th percentile being higher than that at 0.02 MPa(V2) in all three channels. The order of the 75th percentile is A > B > C at all three inlet pressures. The differences between some of these treatments reached significant levels ($p < 0.05$). This indicates that, under the same conditions, channel A has more bubbles with a high Rc-t value, and thus has the best cleaning effect, followed by channel B, and channel C.

### 3.3. Bubble Velocity Vector

The instantaneous velocity vector of a bubble can be obtained by locating the same bubble in two successive frames. Figure 6 shows the velocity vectors of bubble samples obtained for treatments AV1, BV2, and CV3. The direction of the arrow in the figure

represents the direction of the bubble movement at that moment, while the length of the arrow represents the velocity magnitude.

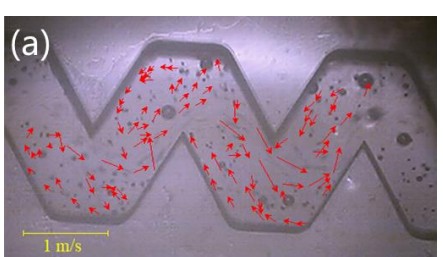 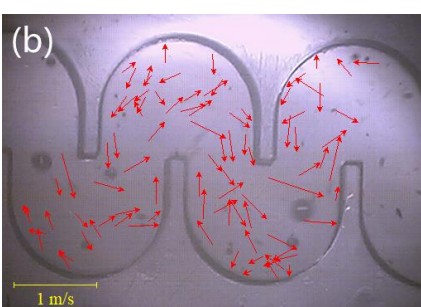 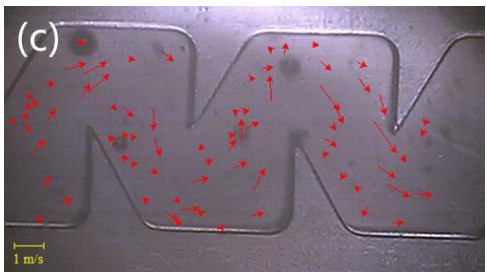

**Figure 6.** An example of the velocity vectors of bubble samples of (**a**) AV1, (**b**) BV2, and (**c**) CV3. The red arrow represents the velocity vector of a bubble.

The direction and magnitude of the instantaneous velocity of the bubbles were clearly affected by their position in the channel and by the inlet pressure. For all the treatments, the direction of the bubble velocity was consistent with the direction of the fluid flow, and the velocity value was large in the mainstream region. In contrast, the velocity value was small in the vortex zone, and the direction of bubble movement close to the inner wall of the channel was opposite to the direction of fluid flow. The maximum instantaneous velocity of the bubble was always measured at the top of the baffles, while the minimum value was always measured at the corner of the groove in the downstream side of the baffles. With increasing inlet pressure, the extent of the mainstream region and the distribution of high-velocity bubbles expands, indicating that the drag force of water on the bubbles is the main influence on their velocity.

The instantaneous velocity values of the bubbles in the 27 experiment groups were extracted, their mean values counted, and their variance analyzed. The results are shown in Figure 7. The mean instantaneous speed of the bubbles in the nine treatments was between 0.214 m/s and 0.488 m/s. Increasing the inlet pressure increased these values for all three channels. The differences between the three different inlet pressures were statistically significant. However, at the same inlet pressure, the mean instantaneous speed showed little difference between the three channels. It can be concluded, therefore, that an increase in inlet pressure results in a significant increase in the instantaneous velocity of the bubbles and affects the extent to which those velocities are distributed.

However, with the same labyrinth channel size, the distribution characteristics of the direction and values of the bubble velocity vector are basically the same. The labyrinth channel geometry has no significant effect on the average value of the bubble instantaneous speed.

The energy dissipation of the labyrinth channel is mainly provided by the turbulence of the water flow in the vortex zone. The movement and coalescence of bubbles in the vortex zone intensifies the energy dissipation. At the same time, the larger the bubble velocity in the vortex zone, the stronger the disturbance on the sediment, which is more conducive to the cleaning of the labyrinth passage. As shown by the results of the bubble trajectories in Section 3.2, the greater the inlet pressure, the greater the beneficial influence of microbubbles on the hydraulic performance and anti-clogging performance of the labyrinth passage. Notably, although the path length is long in the vortex zone at 0.01 MPa (Figure 5), the effect of bubbles on the hydraulic and anti-clogging performance of the labyrinth passage is likely to be less than that at 0.04 MPa. This is because the mean instantaneous speed was the lowest for all three channels at 0.01MPa, as compared with 0.02 and 0.04 MPa (Figure 7), and the instantaneous speed of the bubbles in the vortex zone was lower than the mean value (Figure 6).

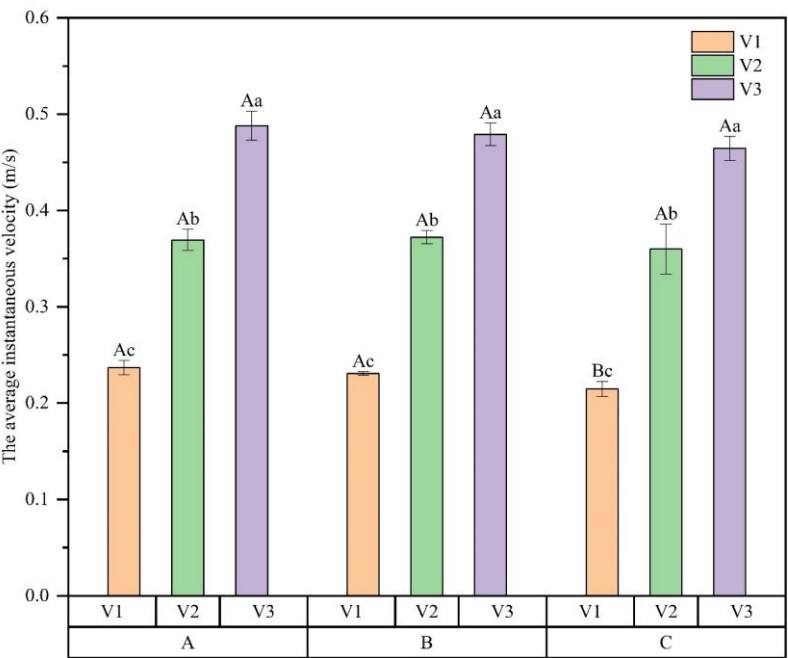

**Figure 7.** The mean instantaneous speed of the bubbles and a difference analysis for the nine treatments. Different uppercase letters indicate a significant difference among different channels at the same inlet pressure; Different lowercase letters indicate significant difference among different inlet pressures for the same labyrinth channel ($p < 0.05$).

## 4. Conclusions

In this paper, the flow pattern of gas–liquid two-phase flow and the movement characteristics of bubbles in the labyrinth channel are described in detail. The conclusions are as follows: (1) The gas–liquid two-phase flow created using micro-nanobubble generating equipment is mainly bubbly flow in the labyrinth channel, but slug flow patterns occasionally emerge. (2) The movement trajectories of the bubbles are smooth in the mainstream region, with a lot of turbulence in the vortex zone. The increasing inlet pressure results in greater numbers of bubbles entering the vortex zone, increased path lengths, and increased mean instantaneous velocity, which are beneficial for cleaning sediment in the vortex zone. (3) The instantaneous velocity distribution of bubbles is consistent in the three channels, with no significant differences at the same pressure. However, the order of path length in the vortex zone is A > B > C, indicating that channel A has the best cleaning effect, followed by channel B, and channel C. This research should help inform the future usage of aerated drip irrigation.

**Author Contributions:** Conceptualization, Z.Z.; Methodology, G.W. and X.Z.; Data curation, W.L.; Writing—original draft, Y.L.; Writing–review & editing, H.L. and B.S. All authors have read and agreed to the published version of the manuscript.

**Funding:** This study was supported by International Programs & Strategic Innovative Programs (grant number: 2022YFE0100300), the National Natural Science Foundation Program (grant number: 51909113, 41901032, 41771256), and the Major Scientific and Technological Innovation Projects of Shandong Key R & D Plan (2019JZZY010710).

**Data Availability Statement:** Not applicable.

**Acknowledgments:** This study was supported by Yantai Key Laboratory of Coastal Hydrological Processes and Environmental Security. We greatly appreciate the help of Chuantao Wang during the field experiment.

**Conflicts of Interest:** The authors declare no conflict of interest.

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
