# Peer review of "Motion Characteristics of Gas–Liquid Two-Phase Flow of Microbubbles in a Labyrinth Channel Used for Aerated Drip Irrigation"

_water, doi:10.3390/w15071432_

Round 1

Reviewer 1 Report

Introduction:

The reviewed paper aims to characterize bubble flows in 3 types of labytrinth channel under 3 different pressures. The introduction of the paper gives good context to the overall importance of analyzing the ADI flow but lacks particular context on the importance of the specific analysis performed in the paper. There is a citation missing:

1.1 - Line 44-46. This sentence needs a citation.

Methods

The experimental setup is mostly well documented. But the data processing techniques and test design is missing important information such as:

2.1 - What was the algorithm used to identify the bubbles between frames?

2.2 - How many bubbles/ bubble trajectories were sampled in the videos?

2.3 - How were these trajectories selected? Was it an automatic selection or was it a manual selection?

2.4 - How does shooting the video at 30/60/90 min influence the results?

Results:

The analysis of flow patterns and velocity is shallow and doesn't give a lot of insight on the performance of these geometries. This section should be revised and the results re-processed so that we can have a better understanding of the influence of these geometries in bubbly flows. Before publishing these points should be addressed:

3.1 - Line 196. Were these 1-2 mm bubbles generated by the machine or created by coalescense of bubbles before the labyrinth structure? Could you clarify it on the text?

3.2 - Figure 5: This picture lacks resolution and the axis labels are difficult to read.

3.3 - Figure 5: Why did the authors opted for a bar plot? The information could be better presented by a scatter plot.

3.4 - The number of looped bubbles seems to be disconnected from a performance analysis. What is the impact on performance of having these looped bubbles? Pressure increase increased the number of loops but there is no connection to the impact on the velocity in subchapter 3.3. There should be a control experiment without bubble generation to infer the impact.

3.5 - Other more relevant characteristics such as bubble residence time should be evaluated. This will allow a better comparison with existing literature and is highly relevant when talking about the clogging performace.

3.6 - Line 301. Multiple bubbles? How many? How were they selected?

Conclusions

The conclusions of the document seem very shallow and have little connection to the given context in the introduction. The authors should give recomendations and comment on the influence  of the geometries tested. Then reflect on the impact of the more complex bubble trajectories that occur on higher pressures.

Reviewer 2 Report

In this paper, the particle tracking technology is used to study the gas-liquid two-phase flow in labyrinth channels with three inlet pressures and different geometric shapes, which is helpful to understand the gas-liquid two-phase flow in labyrinth channels used in aerated drip irrigation, and provide data for the study of the hydraulic performance and anti-clogging performance of aerated drip irrigation systems. This article is intended to be published after minor revisions. The following are some suggestions for changes:

1. Abstract requires certain quantitative results. Abstract part is an important part of this research, which needs better concrete data to highlight the conclusions of the research and ensure the specificity and accuracy of the results. For example, after the inlet pressure increases, the increase proportion of the average number of rings, the increase or decrease of the average circumference of each ring, and so on.

2. Some pictures used in the article need clearer representation. As shown in Figure 1, drawing software (such as cad) can be used to represent channel sizes of different shapes, and ensure that the same positions of channels of different shapes correspond one by one; The bubble trajectories in FIG. 4 and the velocity vectors in FIG. 6 are not clear enough, nor can they well reflect the regularity of the conclusions obtained. The drawing software can be used to draw several or more representative bubble trajectories and vectors in the mainstream and vortex regions, which can better explain the phenomenon of the conclusions.

3. The conclusion fails to well interpret the relationship between the water conservancy performance of drip irrigation system and the resistance to Doucet. The experimental results are relatively ideal, but the conclusions are not closely related to the actual application of drip irrigation system, so relevant descriptions need to be added.

Round 2

Reviewer 1 Report

Thanks for addressing the points mentioned. All of them had satisfying answers.